# Efficiencies of O-MBR and A/O-MBR for Organic Matter Removal from and Trihalomethane Formation Potential Reduction in Domestic Wastewater

**DOI:** 10.3390/membranes12080761

**Published:** 2022-08-02

**Authors:** Sornsiri Sriboonnak, Aegkapan Yanun, Phacharapol Induvesa, Chayakorn Pumas, Kritsana Duangjan, Pharkphum Rakruam, Saoharit Nitayavardhana, Prattakorn Sittisom, Aunnop Wongrueng

**Affiliations:** 1Department of Environmental Engineering, Faculty of Engineering, Chiang Mai University, Chiang Mai 50200, Thailand; sornsiri_sri@cmu.ac.th (S.S.); benentaneer41@gmail.com (A.Y.); pharkphum@eng.cmu.ac.th (P.R.); saoharit@eng.cmu.ac.th (S.N.); prattakorn.s@cmu.ac.th (P.S.); 2Bodhivijjalaya College, Srinakharinwirot University, Nakhon Nayok 26120, Thailand; phacharapol@g.swu.ac.th; 3Research Center in Bioresources for Agriculture, Industry and Medicine, Chiang Mai University, Chiang Mai 50200, Thailand; chayakorn.pumas@gmail.com; 4Department of Biology, Faculty of Science, Chiang Mai University, Chiang Mai 50200, Thailand; 5Science and Technology Research Institute, Chiang Mai University, Chiang Mai 50200, Thailand; kritsana.du@gmail.com

**Keywords:** domestic wastewater, anoxic/oxic membrane bioreactor, oxic membrane bioreactor, trihalomethane formation potential, solid retention time

## Abstract

Lab-scale anoxic/oxic membrane bioreactor (A/O-MBR) and oxic membrane bioreactor (O-MBR) systems using a submerged polysulfone hollow-fiber membrane module with a pore size of 0.01 μm and a total surface area of 1.50 m^2^ were used to treat domestic wastewater. The sludge retention time (SRT) of each system was examined by setting the SRT to 10, 20, and infinity (no sludge withdrawal). The results showed that the total nitrogen removal efficiency of the A/O-MBR was more significant than that of the O-MBR at a SRT of infinity, with figures of 72.3% and 33.1% being found, respectively. The COD removal efficiencies of the A/O-MBR system with a SRT of 10 days, 20 days, and infinity were 82.4%, 84.3%, and 91.5%, respectively. The COD removal efficiencies of the O-MBR system with a SRT of 10 days, 20 days, and infinity were 79.3%, 81.5%, and 89.8%, respectively. An increase in the SRT resulted in an increase in the COD removal efficiency. The FEEM peak of the influent tended to decrease after an increase in the SRT for both systems (A/O-MBR and O-MBR). For the A/O-MBR system, the trihalomethane formation potential (THMFP) was significantly reduced by 88.91% (at a SRT of infinity). The THMFP declined significantly by 85.39% for the O-MBR system at a SRT of infinity. The A/O-MBR system showed a slightly higher efficiency than the O-MBR system in terms of the COD removal and the THMFP reduction. These results indicated that the MBR process, and the A/O-MBR system, in particular, could be used as an effective wastewater treatment process for many developing countries that are troubled by the emerging contamination of water and wastewater.

## 1. Introduction

Water degradation has consistently been a prominent issue and is evident in a number of watershed areas in Thailand. This is due to the continued expansion and development of industries and urban communities, which contribute to qualitative problems involving a variety of organic and inorganic contaminants. These substances are released into water sources through domestic consumption and practices such as bathing, cleaning, cooking, manufacturing, and industrial production. The wastewater in Thailand from hotels, services, and dorms accounts for 67.57%, 27.03%, and 5.40% of the total, respectively [1]. Diverse forms of organic and inorganic substances can be found in these wastewaters. Proteins, carbohydrates, lipids, and nucleic combinations are organic contaminants that can be identified with the chemical oxygen demand (COD) and biological oxygen demand (BOD), whereas inorganic contaminants include ammonia, phosphate, nitrate, and sulfate. A conventional wastewater treatment plant cannot effectively remove natural organic matter (NOM) and/or dissolved organic matter (DOM) with molecules smaller than 0.45 μm [2]. DOM compounds such as humic acid and fulvic acid contaminate the water. NOM is produced when plant and animal matter decompose and form a complex combination of organic molecules. Chlorine, chloramines, ozone, and chlorine dioxide are the most frequent disinfectants used for disinfection [3]. These chemicals react to NOM and/or DOM to produce mutagenic and carcinogenic disinfection by-products (DBPs) [4,5,6]. Currently, there is a lack of water management and measures to prevent the entry of wastewater emissions into water resources. Therefore, it is essential to find solutions to these problems. Such an approach must be effective and sustainable. Membrane bioreactors (MBRs) have recently become popular for treating municipal and industrial wastewater [7]. The MBR process has many remarkable benefits over the conventional activated sludge system, including a higher biomass concentration, less sludge production, and high-quality effluent. As a result, membrane bioreactors have been developed to eliminate the limitations of conventional treatment systems [8]. Membrane bioreactors can handle more organic loads than conventional treatment methods. Membrane bioreactors can treat wastewater containing specific components that are hazardous to microbial function. They are resistant to environmental changes during treatment since their microbial masses are long-lasting. Additionally, membrane bioreactors improve the treated effluent’s quality [9,10]. MBRs have been effectively combined with anoxic/oxic (AO) processes to enhance the removal of both organics and nutrients. Recently, Adoonsook D. et al. [11] studied a simplified technique for the simultaneous removal of nitrogen and phosphorus using the A/O-MBR system by merging biofilms into anaerobic compartments containing active biomass. Moreover, studies on nutrient removal using MBRs utilizing various carbon sources have focused on evaluating the microbial population and its compositional responsibilities [12]. Liu et al. [13] demonstrated that a two-stage AO-MBR system was beneficial for pollutant removal in landfill leachate, and the average removal efficiencies of the chemical oxygen demand (COD) and the total nitrogen (TN) were 80.6% and 74.9%, respectively. In addition, the membrane bioreactor can minimize chlorine usage during post-treatment effluent disinfection. For the disinfection procedure, chlorine gas, chlorine dioxide, and chloramine are commonly added to the effluent. However, these disinfectants can react with natural organic matter in the effluent and form disinfection by-products (DBPs) [14]. DBPs affect the human body, as studies have shown that these substances are carcinogenic and cause mutations in laboratory animals [5,6,14,15]. Most THM precursors are created when dissolved organic matter is present, which is responsible for the THM formation potential (THMFP). The trihalomethane formation potential (THMFP) of raw or treated water sources indicates the highest trihalomethane (THM) levels that are likely to occur when chlorine reacts with the THM precursors in water [16,17]. This study focuses on trihalomethane (THM) precursors and the trihalomethane formation potential (THMFP). To date, there has not been an overview of A/O-MBR and O-MBR technology employed to remove THM precursors and THMFP; hence, this technology is extensively discussed in this paper. This research aims to study the efficiencies of oxic and anoxic/oxic membrane bioreactors (O-MBR and A/O-MBR) on the organic matter removal from and trihalomethane formation potential reduction in domestic wastewater. The effect of sludge age on these factors is investigated by varying the sludge retention time (SRT) between 10 days, 20 days, and infinity (no sludge withdrawal).

## 2. Materials and Methods

### 2.1. Domestic Wastewater

The domestic wastewater used in this study came from a wastewater treatment plant located in an educational institute in Chiang Mai, Thailand, which has a capacity of 8500 m^3^/day. The sample was collected before entering the activated sludge aeration tank. The sampling frequency was twice a week, with a grab sample being taken. It was collected in a clean plastic container. The seed used to start the system was collected from the aeration tank of the wastewater treatment plant in the educational institute.

### 2.2. Anoxic/Oxic and Oxic Membrane Bioreactors (A/O-MBR and O-MBR)

This study was carried out using two lab-scale anoxic/oxic and oxic membrane bioreactors (A/O-MBR and O-MBR). The reactors were operated in parallel and fed with domestic wastewater. The A/O-MBR and O-MBR systems used in this experiment are illustrated in Figure 1 and Figure 2, respectively. In the aerobic MBR tank of each system, a polysulfone (PS) hollow-fiber membrane module with a pore size of 0.01 μm and a total membrane area of 1.50 m^2^ was installed. Air diffusers were constructed underneath the membrane modules to continuously supply oxygen for biomass growth. A water level controller was installed for each system to maintain a constant water level corresponding to a total hydraulic retention time (HRT) of 10 h. A peristaltic pump was used to introduce domestic wastewater from the feed tank to both systems. A return pump was used for recirculating the sludge. The sludge retention time (SRT) of each system was studied by varying the sludge retention time between 10 days, 20 days, and infinity (no sludge withdrawal). A permeate pump was used to pull water through the polysulfone (PS) hollow-fiber membrane. A differential pressure gauge was used to measure the transmembrane pressure (TMP) of the membrane module. The transmembrane pressure (TMP) was controlled at 30 kPa in all experiments. In addition, when the permeate production was fluctuated, the membrane was backwashed. The initial mixed liquor suspended solids (MLSSs) concentration was approximately 3000 mg/L.

### 2.3. Analytical Methods

The mixed liquor suspended solids (MLSSs) and mixed liquor volatile suspended solids (MLVSSs) were analyzed according to the standard methods 2540D and 2540E, respectively [18]. For the MLSS analysis, the samples were dried for a minimum of 2 h in a 103–105 °C oven. For the MLVSS analysis, the samples were ignited at 550 °C. The remaining solids were then fixed (inorganic), and the ignition loss was the volatile (organic) solids level. The chemical oxygen demand (COD) and total nitrogen (TN) were calculated using the summation of the total Kjeldahl nitrogen (TKN), nitrite (${\mathrm{NO}}_{2}^{-}$), and nitrate (${\mathrm{NO}}_{3}^{-}$) concentrations (this formula was used for the measurement of nitrogen in wastewater), which were measured according to the standard methods for the examination of water and wastewater [18]. Dissolved oxygen (DO) and pH were determined using a WTW Multi pH/Oxi 340i (WTW GmbH, Weilheim, Germany) dissolved oxygen meter. The fluorescence excitation–emission matrix (FEEM) was analyzed to obtain the THMFP precursor characteristics of the raw wastewater and permeate. The excitation wavelength ranged from 220 nm to 600 nm and increased at intervals of 5 nm. The FEEM was measured using a spectrofluorometer (JASCO, FP-6200, Bangkok, Thailand). The water samples were filtered through a 0.7 µm GF/F filter and 0.45 μm nylon membrane filter, respectively, prior to measurements. Prior to use, the GF/F filters were combusted at 550 °C to avoid organic matter contamination. The nylon membrane filters were rinsed with 100 mL of pure water followed by a 50 mL water sample prior to use.

### 2.4. Statistical Analysis

All results were analyzed using a one-way ANOVA at a 95% confidence level. At *p* > 0.050, the result was insignificant; when *p* < 0.050, the result was significant.

### 2.5. Trihalomethane Formation Potential (THMFP) and Analysis

In the case of the THMFP measurement, the raw wastewater and treated wastewater samples were examined following the standard methods 5710 B, 4500-Cl B, and 6232 B [18]. The formation potential of the THMs was analyzed for chloroform (CHCl_3_), bromodichloromethane (CHBrCl_2_), dibromochloromethane (CHBr_2_Cl), and bromoform (CHBr_3_). The THMFP measurement consisted of three steps: (1) a free chlorine residual measurement, (2) liquid–liquid extraction, and (3) sample analysis. Firstly, the THMFP test was conducted for 7 days. At the end of the 7-day reaction period, samples had a remaining free chlorine residual of 3–5 mg/L. The residual chlorine was measured using a portable spectrophotometer (HACH, DR/890 colorimeter) before proceeding to the next step. Pentene was used as the solvent. The extracted solution was kept in a refrigerator under 4 °C. Finally, the extracted solution was taken and analyzed using gas chromatography (GC-ECD) (Hewlett Packard, HP 6890 GC, Agilent Technologies, Inc., Santa Clara, CA, USA) with an RTX-624 column.

## 3. Results and Discussion

### 3.1. Domestic Wastewater Characteristics

The domestic wastewater characteristics are reported in Table 1. The concentration of suspended solids (SS) was 268.0 ± 172.2 mg/L, and the SS during the experiment is shown in Appendix A. Additionally, the biological oxygen demand (BOD) was 116.7 ± 12.4 mg/L. It was found that these values were above the effluent standards of 30 mg/L and 20 mg/L, respectively [1]. Due to the amount of septic tank wastewater and effluent from apartments, offices, and university buildings, including wastewater from hospitals, which contains organic materials, the SS and BOD values increased due to higher standards and higher values. According to a pollution control department report, the wastewater from these dormitories had SS and BOD values of 695 mg/L and 798 mg/L, respectively. The SS and BOD values in the hospital wastewater were 87 mg/L and 238 mg/L, respectively.

### 3.2. Efficiencies of the A/O-MBR and O-MBR Systems

#### 3.2.1. Chemical Oxygen Demand

Figure 3 and Figure 4 illustrate how the COD concentrations in the A/O-MBR and O-MBR systems changed during the experiment at different SRTs. Table 2 shows the COD concentrations at different SRTs in the A/O-MBR and O-MBR systems. In the A/O-MBR system, the COD concentration in the raw water influent fluctuated. However, the COD concentrations in the permeate from the A/O-MBR system were stable. The COD concentration in the raw water influent for SRTs of 10 days, 20 days, and infinity were 193.5 ± 85.3 mg/L, 174.1 ± 80.6 mg/L, and 218.6 ± 61.1 mg/L, respectively. The COD concentrations in the permeate for SRTs of 10 days, 20 days, and infinity at steady state were 28.9 ± 8.9 mg/L, 28.9 ± 5.1 mg/L, and 17.4 ± 4.1 mg/L, respectively. Since the two systems were operated in parallel, the COD concentrations in the raw water influent in the O-MBR system were similar. The COD concentrations in the permeate for SRTs of 10 days, 20 days, and infinity at steady state were 29.3 ± 10.7 mg/L, 30.5 ± 3.7 mg/L, and 21.2 ± 4.2 mg/L, respectively. In addition, the effects of the pH [1,19,20], alkalinity [21,22], dissolved oxygen [23,24], and the ratio of MLVSSs to MLSSs [25], supporting the COD results, are shown in the Appendix A.

The effects of the A/O-MBR and O-MBR systems in terms of COD removal are shown in Figure 5. The COD removal efficiencies of the A/O-MBR (System 1) and O-MBR (System 2) systems at a SRT of 10 days were 82.4% and 79.3%, respectively. The COD removal efficiencies of Systems 1 and 2 at a SRT of 20 days were 84.3% and 81.5%, respectively. The COD removal efficiencies of Systems 1 and 2 at a SRT of infinity were 91.5% and 89.8%, respectively. It was observed that an increase in the SRT led to an increase in the COD removal efficiency. The reduction in the COD was due to the conversion of organic matter into carbon dioxide, water, and microbial cells during the decomposition of organic matter [26]. The membrane bioreactor systems disposed of excess sludge more efficiently than conventional treatment systems. As a result, the amount of organic matter used in cell production was lower. Therefore, the conversion of organic matter into carbon dioxide was responsible for most of the COD loss.

#### 3.2.2. Effect of Nitrification/Denitrification 

Figure 6 and Figure 7 illustrate the changes in the total nitrogen concentrations during experiments at different SRTs in the A/O-MBR and O-MBR systems, respectively. Meanwhile, Table 3 demonstrates the total nitrogen concentrations at different SRTs in the A/O-MBR and O-MBR systems. In the A/O-MBR system, the total nitrogen concentration in the raw water influent fluctuated. However, the total nitrogen concentrations in the permeate from the A/O-MBR system were stable. The total nitrogen concentration in the raw water influent for SRTs of 10 days, 20 days, and infinity were 35.5 ± 2.9 mg/L, 31.6 ± 4.6 mg/L, and 32.4 ± 3.1 mg/L, respectively. The total nitrogen concentrations in the permeate for SRTs of 10 days, 20 days, and infinity at the steady state were 22.0 ± 2.0 mg/L, 12.7 ± 0.9 mg/L, and 8.9 ± 0.6 mg/L, respectively. Since the two systems were operated in parallel, the total nitrogen concentrations in the raw water influent in the O-MBR system were similar. The total nitrogen concentrations in the permeate for SRTs of 10 days, 20 days, and infinity at the steady state were 29.0 ± 2.6 mg/L, 23.9 ± 2.5 mg/L, and 21.4 ± 1.6 mg/L, respectively. An increase in the SRT resulted in an increase in the TN removal efficiency.

Interestingly, in terms of the nitrification process, NH_3_ (in terms of the TKN) in both the A/O-MBR and O-MBR was decreased in all SRTs (Appendix A). The immediate product (${\mathrm{NO}}_{2}^{-}$) (Appendix A) showed a similar trend to the TKN. In the final product of the nitrification process, ${\mathrm{NO}}_{3}^{-}$ (Appendix A), however, showed a different trend between the A/O-MBR and O-MBR. Because of the presence of oxygen, the ${\mathrm{NO}}_{3}^{-}$ concentration in the O-MBR (aerobic condition) was higher than in the A/O-MBR (anoxic condition). To confirm the denitrification process, which is the process that converts nitrate to nitrogen gas in the absence of oxygen, the nitrate concentration in the permeate from the A/O-MBR showed a lower concentration than that obtained from the O-MBR. This could be confirmed by the occurrence of the denitrification process.

The efficiencies of the A/O-MBR and O-MBR systems in terms of the total nitrogen removal are shown in Figure 5**.** At a SRT of 10 days, the total nitrogen removal efficiencies of the A/O-MBR (System 1) and O-MBR (System 2) systems reached 38.1% and 19.0%, respectively. At a SRT of 20 days, the total nitrogen removal efficiencies of Systems 1 and 2 were 59.7% and 23.4%, respectively. At a SRT of infinity, the total nitrogen removal efficiencies of Systems 1 and 2 were 72.2% and 33.1%, respectively. It was noted that a rise in the SRT resulted in an increase in the total nitrogen removal efficiency.

#### 3.2.3. FEEM

The FEEM is a technique for identifying natural organic substances in water. Chen et al. [27] described the extent of excitation (Ex) and emission (Em) wavelengths according to five sections, namely, tyrosine, aromatic proteins, fluvic acid, microbial by-product-like, and humic acid substances. Their Ex/Em FEEM peaks occurred at 220–250 nm/280–330 nm, 220–250 nm/330–380 nm, 220–280 nm/>380 nm, >250 nm/280–330 nm, and >250 nm/>380 nm, respectively. The FEEM in the domestic wastewater and permeated for A/O-MBR and O-MBR systems are reported in Figure 8, Figure 9 and Figure 10. The Ex/Em FEEM peaks occurred at 340–355 nm/410–435 nm and 275–290 nm/345–425 nm, respectively. These peaks were designated as A and B, respectively. According to past reports, peaks A and B corresponded to humic acid and fulvic acid [28], respectively. All the wastewater influents exhibited peaks A and B at SRTs of 10 days, 20 days, and infinity. In both the A/O-MBR and O-MBR systems, the intensities of the FEEM peaks A and B in the permeate were lower than those in the wastewater influent at all SRTs. The FEEM intensity in both systems also declined as the SRT increased. The FEEM intensity of the A/O-MBR system showed a higher efficiency than that of the O-MBR system. Our findings were consistent with the COD removal efficiency of both systems under all SRTs. Figure 5 shows a rise in the COD removal efficiency of both systems with an increase in the SRT as the organic matter was transformed into carbon dioxide, water, and microbial cells. From the FEEM and COD concentrations, it was concluded that the A/O-MBR system had good efficiency. Thus, the MBR system might minimize the amount of organic matter in wastewater and decrease the generation of THMFP.

#### 3.2.4. TTHMFPs

The measured THM concentration was obtained after the potential formation test in all the experiments. Figure 11 shows that water intake was observed at a SRT of 10 days. Three THMs (${\mathrm{CHCl}}_{3}$, CHBrCl_2_, and CHBr_2_Cl) had a formation potential of 114 and 6213 μg/L in the raw wastewater. CHBr_3_ was not detected in the raw wastewater, and it permeated both systems. Chloroform was determined to be the most common THM species. The formation potential of the four THMs slightly decreased in the anoxic/oxic tank water (System 1) (range of 52 to 2615 μg/L) when compared to the oxic tank water (System 2) (range of 103 to 5518 μg/L). Aside from that, the range of the four THMFPs at a SRT of 20 days was 52–7063 μg/L in the raw wastewater. Chloroform formation was still the highest in this system, even though CHBr_3_ was not detected. Furthermore, when compared to anoxic/oxic tanks, similarities between the THMFP and oxic tank water (System 2) were discovered. In the anoxic/oxic tank (System 1) at a SRT of infinity, chloroform presented a similar trend with both systems (SRTs of 10 and 20 days). The range of the THMFP was 193–7005 μg/L in the raw wastewater. When considering the water outlet in the oxic tank water (System 2) at a SRT of infinity, the formation potential showed a lower concentration (range of 38–1027 μg/L) compared with the anoxic/oxic tank at a SRT of infinity. According to these results, the formation potential of total trihalomethanes (TTHMFPs) in the permeate was much lower with the A/O-MBR system than with the O-MBR system at a SRT of infinity. As shown in Figure 5, the A/O-MBR system had an increased COD removal efficiency value, which suggested a decreased precursor to the creation of TTHMFPs. Additionally, the COD removal efficiency value illustrated an enhanced precursor to the development of TTHMFPs. The following conclusions may be drawn from the findings: the intensities of FEEM peaks A and B in the permeate were lower when contrasted with those of the wastewater influent for both systems at all SRTs. Figure 8, Figure 9 and Figure 10 present the FEEM analysis findings on the intensities. Four TTHMFP species, ${\mathrm{CHCl}}_{3}$, CHBrCl_2_, CHBr_2_Cl, and CHBr_3_, were detected in the domestic wastewater results shown in Figure 11. In this study, the predominant species was ${\mathrm{CHCl}}_{3}$. Sriboonnak S. et al. [29] found that the dominant TTHM species in the raw water samples obtained from a surface water reservoir and water distribution networks was ${\mathrm{CHCl}}_{3}$. Furthermore, when considering leachate-contaminated groundwater, the formation of CHCl_3_ was the most prevalent among the TTHMFPs [30].

## 4. Conclusions

The findings of the experiments demonstrated that the A/O-MBR and O-MBR systems were both very successful at treating domestic wastewater. The sludge retention time (SRT) of each system was investigated by adjusting the SRT to 10, 20, and indefinite intervals (no sludge withdrawal). An increase in the SRT led to a corresponding increase in the efficiency with which the COD was removed from both systems. Both the A/O-MBR and the O-MBR systems were quite successful in disposing of suspended particles as well as the total nitrogen. In the A/O-MBR system, the trihalomethane formation potential (THMFP) was cut down by a significant amount (88.91%) at a SRT of infinity. In addition, the THMFP levels dramatically declined for the O-MBR system (85.39%) at a SRT of infinity. Regarding the elimination of the COD and the lowering of the THMFP, the A/O-MBR system revealed a somewhat greater level of efficiency than the O-MBR system. 

## 5. Patents

This research has no patents.

## Figures and Tables

**Figure 1 membranes-12-00761-f001:**
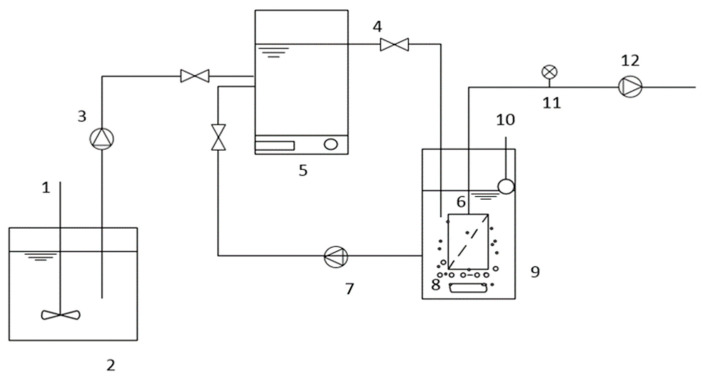
The anoxic/oxic membrane bioreactor (A/O-MBR) system (System 1). The system consisted of (1) an agitator; (2) a feed tank; (3) a feed pump; (4) an anoxic tank; (5) a magnetic stirrer; (6) a membrane; (7) a return pump; (8) an air diffuser; (9) an aerobic tank; (10) a water level controller; (11) a pressure gauge; and (12) a permeate pump.

**Figure 2 membranes-12-00761-f002:**
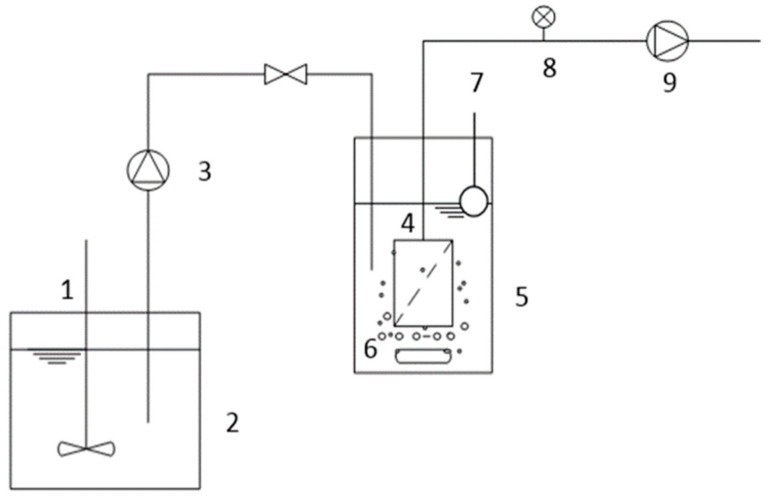
The oxic membrane bioreactor (O-MBR) system (System 2). The system consisted of (1) an agitator; (2) a feed tank; (3) a feed pump; (4) a membrane; (5) an aerobic tank; (6) an air diffuser; (7) a water level controller; (8) a pressure gauge; and (9) a permeate pump.

**Figure 3 membranes-12-00761-f003:**
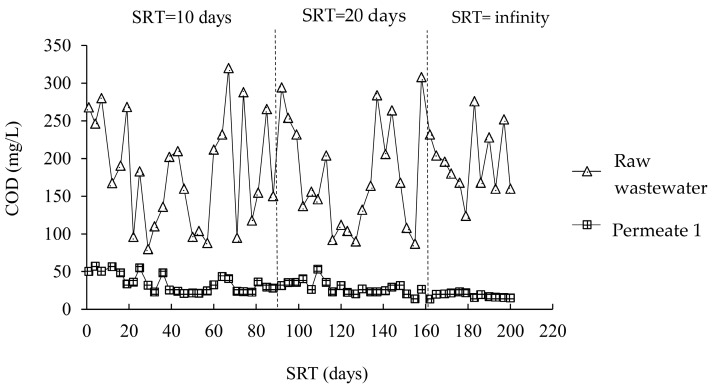
COD concentrations during the experiment at different SRTs in the A/O-MBR system (System 1).

**Figure 4 membranes-12-00761-f004:**
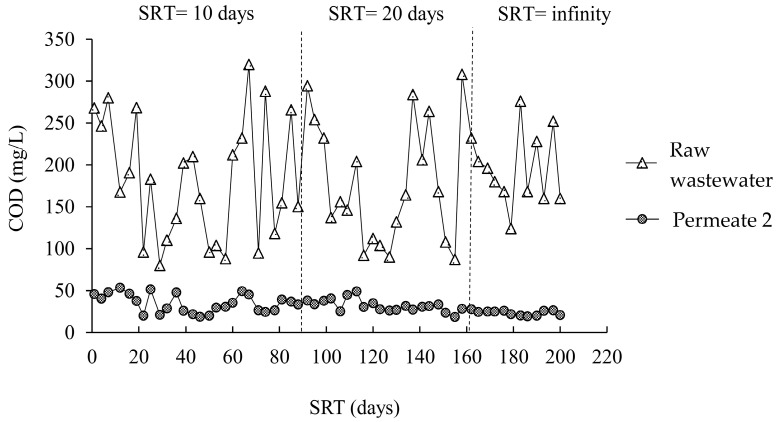
COD concentrations during the experiment at different SRTs in the O-MBR system (System 2).

**Figure 5 membranes-12-00761-f005:**
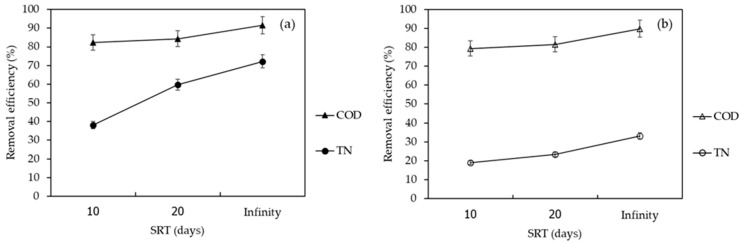
COD and TN removal efficiency at different SRTs in (**a**) the A/O-MBR system (System 1) and (**b**) the O-MBR system (System 2).

**Figure 6 membranes-12-00761-f006:**
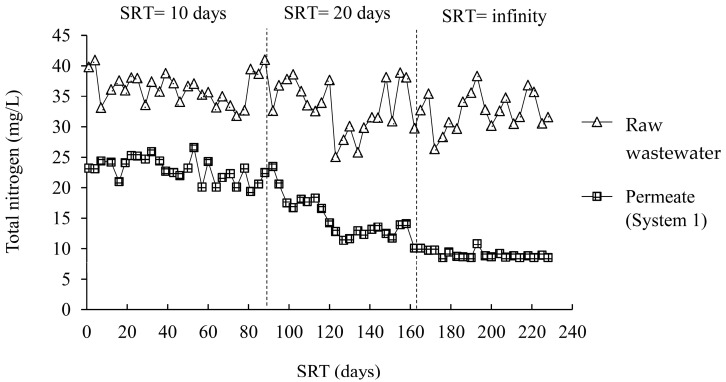
Total nitrogen concentrations during the experiment at different SRTs in the A/O-MBR system (System 1).

**Figure 7 membranes-12-00761-f007:**
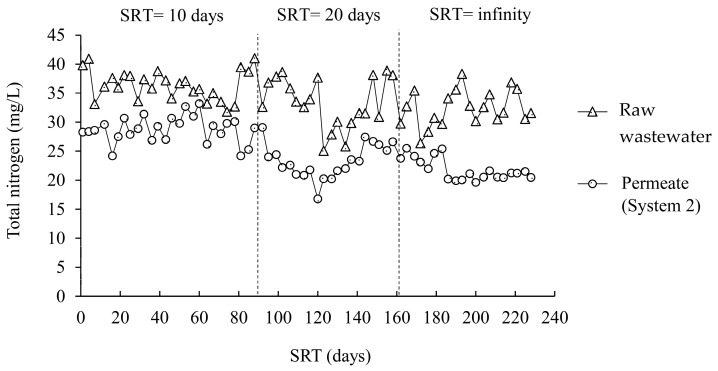
Total nitrogen concentrations during the experiment at different SRTs in the O-MBR system (System 2).

**Figure 8 membranes-12-00761-f008:**
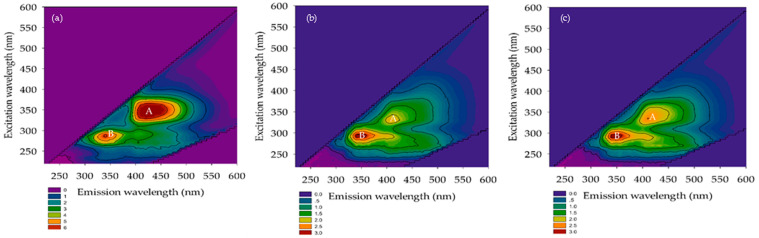
FEEM of the A/O-MBR and O-MBR systems at a SRT of 10 days. (**a**) Raw wastewater. (**b**) Permeate from A/O-MBR (System 1). (**c**) Permeate from O-MBR (System 2).

**Figure 9 membranes-12-00761-f009:**
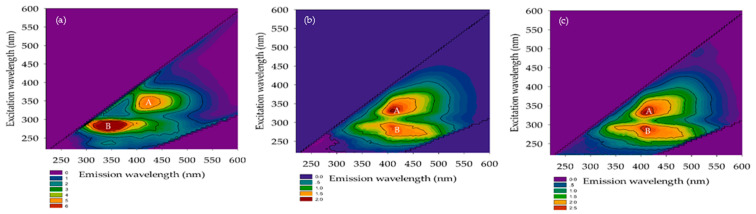
FEEM of the A/O-MBR and O-MBR systems at a SRT of 20 days. (**a**) Raw wastewater. (**b**) Permeate from A/O-MBR (System 1). (**c**) Permeate from O-MBR (System 2).

**Figure 10 membranes-12-00761-f010:**
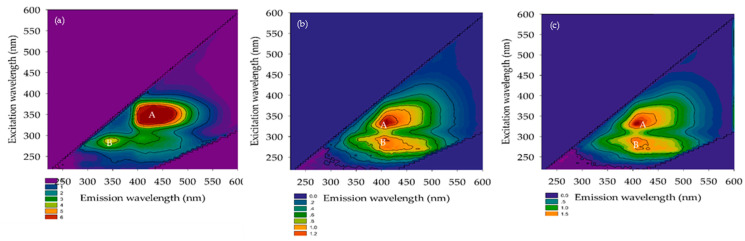
FEEM of the A/O-MBR and O-MBR systems at a SRT of infinity. (**a**) Raw wastewater. (**b**) Permeate from A/O-MBR (System 1). (**c**) Permeate from O-MBR (System 2).

**Figure 11 membranes-12-00761-f011:**
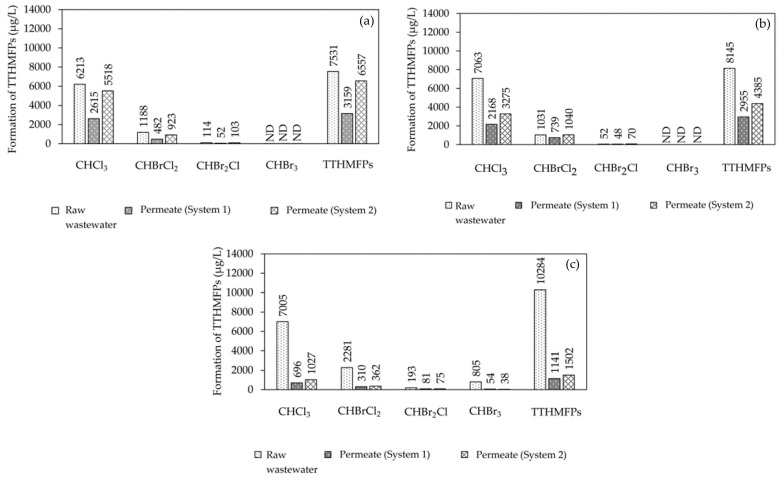
The formation of TTHMFPs at (**a**) SRT of 10 days, (**b**) SRT of 20 days and, (**c**) SRT of infinity.

**Table 1 membranes-12-00761-t001:** Domestic wastewater characteristics.

Parameters	Range	Value (n = 5)
pH	7.14–7.98	7.51 ± 0.1
SS (mg/L)	50–920	268.0 ± 172.2
Temperature (°C)	20.4–33.8	28.2 ± 3.1
Alkalinity (mg/L as CaCO_3_)	86–176	128.5 ± 20.2
COD (mg/L)	80–308	191.1 ± 69.1
BOD (mg/L)	98.3–133.8	116.7 ± 12.4
TKN (mg/L) Ammonia (mg/L)	24.6–39.4 18.4–32.8	33.5 ± 3.4 24.0 ± 3.2
Nitrite (mg/L)	0.0–1.8	0.4 ± 0.4
Nitrate (mg/L)	0.0–1.4	0.3 ± 0.3
Phosphorus (mg/L)	1.21–1.78	1.5 ± 0.2

**Table 2 membranes-12-00761-t002:** Effects of the A/O-MBR and O-MBR systems on COD removal.

SRT (days)	COD (mg/L)		
	Raw Wastewater	Permeate from A/O-MBR (System 1)	Permeate from O-MBR (System 2)
10	193.5 ± 85.3	28.9 ± 8.9	29.3 ± 10.7
20	174.1 ± 80.6	28.9 ± 5.1	30.5 ± 3.7
Infinity	218.6 ± 61.1	17.4 ± 4.1	21.2 ± 4.2
*p*-value	0.301	0.000	0.000

**Table 3 membranes-12-00761-t003:** Efficiencies of the A/O-MBR and O-MBR systems on total nitrogen removal.

SRT (days)	Total Nitrogen (mg/L)		
	Raw Wastewater	Permeate from A/O-MBR (System 1)	Permeate from O-MBR (System 2)
10	35.5 ± 2.9	22.0 ± 2.0	29.0 ± 2.6
20	31.6 ± 4.6	12.7 ± 0.9	23.9 ± 2.5
Infinity	32.4 ± 3.1	8.9 ± 0.6	21.4 ± 1.6
*p*-value	0.030	0.000	0.000

## Data Availability

Not applicable.

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
