# Peer review of "Efficiencies of O-MBR and A/O-MBR for Organic Matter Removal from and Trihalomethane Formation Potential Reduction in Domestic Wastewater"

_membranes, 2022, doi:10.3390/membranes12080761_

Round 1
Reviewer 1 Report
English of this paper is very poor and need to be revised thoroughly. There are a lot of grammatical mistakes. Most of the sentences are wrong.
I have not gone through the whole paper. I think it’s not good idea to read the whole paper as its written badly. I shall not recommend the paper publication in its current form. It really needs to re-write.
Comment 1: Abstract , line 19: Laboratory should be Lab-scale
Comment 2: abstract, line 23; “The concentration in the wastewater” concentration of what?
Comment 3: Introduction, “The degradation of water resources in Thailand has been around since the past and has been shown in some watershed areas”
Very poor writing. Please improve throughout the paper.
Comment 4: line 48: (DOM) with molecules smaller than 0.45 m???
o.45 m???
Comment 5: line 81: “Domestic wastewater from wastewater treatment plant located in the educational institute, Chiang Mai, Thailand, has a capacity of 8,500 m3 /day”
Sentence is not complete, and this type of writing is almost everywhere.
Comment 6: Results like pH, DO, MLSS and alkalinity are explained unnecessarily. These results can be described in 2 to 3 line complementing other results, not as a separate section.
Comment 7: Figure 10 should be coloured
Reviewer 2 Report
Researchers systematically studied the influence of sludge retention time on the removal of organic matter and reduction of trihalomethane formation from domestic wastewater. However, there are lack of convincing explanations on the following questions.
1. In the introduction section, what is the most updated research on O-MBR and A/O-MBR as well as how sludge age influences the organic matter removal? What is the gap between the recent peer work and this study? What’s the novelty of this study?
2. Why author focus on the “trihalomethane formation potential reduction”?
3. Why the overall TN removal efficiency of A/O-MBR system was lower than the O-MBR system?
4. In the “analytical methods” section, it will be great to list the analytical method and instrument used for mixed liquor suspended solids (MLSS), mixed liquor volatile suspended solids (MLVSS), chemical oxygen demand (COD), total Kjeldahl nitrogen (TKN), ammonia nitrogen (NH4+), nitrite (NO2-), nitrate (NO3-) concentrations characterization.
5. What is the correlation between THM, TTHM, TTHMFP, and THMFP? Could author clarify and keep the abbreviation consistent?
6. What is the explanation for “When it came to the elimination of COD and the lowering of THMFP, the A/O-MBR system demonstrated a somewhat greater level of efficiency than the O-MBR system?”
Round 2
Reviewer 1 Report
Comment 1: Authors have tried to improve the English. However, there are still many grammatical mistakes. I shall suggest authors to send manuscript to native English speaker for proofread.
For example, “The sludge retention time (SRT) of each system was examined by setting the SRT at 10, 20, and forever (no sludge withdrawal)”. Please rephrase it in correct english
Comment 2, abstract: “The concentration of COD in the wastewater influent for SRTs 10 days, 20 days, and infinity were 193.5±85.3 mg/L, 174.1±80.6 mg/L, and 218.2±61.1 mg/L, respectively”
Not a good idea to mention influent concentrations in abstract. Please remove it. Abstract should contain a brief aims of the study, methods, and important results.
Comment 3, abstract: “With A/O-MBR system, trihalomethane formation potential (THMFP) was significantly reduced from 10,284 to 1,141 μg/L (SRT infinity). For O-MBR system, THMFP was also significantly reduced from 10,284 to 1,502 μg/L (SRT infinity)”
Is 10, 284 μg/L is the concentration of THMFP in influent? Please rephrase the sentence to make it more clear and grammatically correct.
Comment 4, abstract: please add TKN, nitrite, nitrate, ammonia and Phosphorus removal efficiencies of both systems in the abstract as well. Add FEEM comparative results in abstract as well.
Comment 5, Introduction: “The conventional wastewater treatment plant (sediment tank) cannot effectively remove natural organic matter (NOM)and/or dissolved organic matter (DOM) with molecules smaller than 0.45 μm” ????
Sedimentation tank is not a conventional wastewater treatment plant.
Comment 6, Introduction: “A complex combination of organic molecules named NOM is produced when plants and animal matters decay”.
Please rephrase
Comment 7, Introduction: “Currently, there is a lack of water management and wastewater emissions to water resources.”????
“Therefore, it is essential to find solutions to these problems. Such an approach must be economically cost-effective and sustainable.”??
Please rephrase the sentence. After writing a sentence, please read it carefully and think if it is making sense
Comment 8, Introduction: “Membrane technology has been implemented in wastewater treatment [7]”. Why repeating it?
Comment 9, Introduction: “The subsequent development of membrane bioreactors addresses the limits of current treatment systems”??? please rephrase
Comment 10, Introduction: “The membrane bioreactor can treat wastewater containing specific components hazardous to microbial function. It is resistant to environmental changes during treatment since its microbial mass is long-lasting. Additionally, the membrane bioreactor improves the efficacy of the treated effluent's quality [9-10].” MBRs have been effectively combined with anoxic-oxic (AO) processes in enhancing the removal of both organics and nutrients. Recently, Adoonsook D. et al. [11] studied a simplified technique for the simultaneous removal of nitrogen and phosphorus in the A/O-MBR system by merging biofilms into anaerobic compartments containing active biomass”???
“Liu et al. [13] demonstrated that a two-stage AO-MBR system was beneficial to pollutant removal in landfill leachate, and the average removal efficiencies of chemical oxygen demand (COD) and total nitrogen (TN) were 80.6% and 74.9%, respectively”
Poor writing style and many grammatical mistakes
Comment 11, Introduction: “Moreover, studies of nutrients in”
“Moreover, studies of nutrients removal in”
Comment 12, Introduction: “This study focuses on trihalomethane (THM) precursors and the trihalomethane formation potential (THMFP). THMFP is the difference between the total THM concentration measured after the chlorination process without treatment and the total THM concentration measured at regular intervals during water treatment”
No one can understand the THMFP definition. Please rephrase it
Comment 13, Introduction: To date, there has not been an overview of the A/O-MBR and O-MBR technology employed to remove THM precursors and THMFP; hence, this technology is extensively discussed in this paper. The research aims to study the efficiencies of oxic and anoxic-oxic membrane bioreactors (O-MBR and A/O-MBR) on organic matter removal and trihalomethane formation potential reduction from domestic wastewater. The effect of sludge age on those was investigated by varying the sludge retention time (SRT) from 10 days, 20 days, to infinity (no sludge withdrawal).
Very poor writing style with a lot of grammatical mistakes. Please rewrite it.
Comment 14, Section 2.3: “Mixed liquor suspended solids (MLSS) were dried for a minimum of 2 hours in a 103 – 105 °C oven. Mixed liquor volatile suspended solids (MLVSS) were ignited at 550°C. The remaining solids are the fixed (inorganic) solids, the ignition loss is the volatile (organic) solids”
“Please Rephrase it”. Cite the standard method used.
Comment 15, Section 2.3: “Fluorescence excitation-emission matrix (FEEM) was measured by a spectrofluorometer (JASCO, FP-6200).”
Provide details of the FEEM method.
Comment 16, Section 2.4: “All results were analyzed at a 95% confidence level. At p>0.050, the result was insignificant; …..”
Please write the test name e.g. t-test???? And rephrase the whole 2.4 section
Comment 17: How you did chlorination of influent and effluent?? Whole method of chlorination is missing. Section 2.5 explains the measurement of chlorination by-products only.
Comment 18: section 3.1 and 3.2.1; please combine these 2 sections and should not be more than 2 paragraphs.
Comment 19: Please add nitrate, nitrite, ammonia phosphate results in results and discussion section and explain it. Authors have mentioned that these parameters were measure but no results and discussion is presented in the paper regarding it. Please be consistent to use total kjeldhal nitrogen “TKN” or total nitrogen throughout the manuscript.
Comment 20, Section 3.2.7: very less information about the peaks is given here. Please explain the differences of both systems here infer the results and add discussion around it. Authors can discuss different types of DOM components from the data they have
Comment 21: please make a separate section on the results of THMs and TTHMFPs and add discussion around it before the conclusion section.
Comment 22: conclusion needed to be rephrased and should add information about the ammonia, nitrite, nitrate, and phosphate removal as well
